# Growth and Diversity of Spoiling and Foodborne Bacteria in Poultry Hamburgers in Modified Atmosphere and with Sulfites During Shelf Life

**DOI:** 10.3390/microorganisms13040754

**Published:** 2025-03-26

**Authors:** Elena González-Fandos, Alba Martínez Laorden, Santiago Condón Usón, María Jesús Serrano Andrés

**Affiliations:** 1Department of Food Technology, CIVA Research Center, University of La Rioja, de la Paz Avenue, 26006 Logroño, Spain; elena.gonzalez@unirioja.es (E.G.-F.); alba.martinezl@unirioja.es (A.M.L.); 2Food Science and Technology Department, Instituto Agroalimentario de Aragón IA2, Universidad de Zaragoza, Miguel Servet St. 177, 50013 Zaragoza, Spain; scondon@unizar.es; 3Instituto Agroalimentario de Aragón IA2, Universidad de Zaragoza-Centro de Investigación y Tecnología Agroalimentaria de Aragón (CITA), Miguel Servet St. 177, 50013 Zaragoza, Spain

**Keywords:** meat products, bacterial identification, MALDI-TOF, decay

## Abstract

Poultry meat is the most consumed meat worldwide due to its low fat content, sensory qualities, and affordability. However, its rapid spoilage, especially when minced for products like hamburgers, is a challenge. Strategies such as sulfite addition or modified-atmosphere packaging (MAP) can help control spoilage and microbial growth. This study evaluated both approaches by analyzing bacterial development in poultry hamburgers through total viable counts and MALDI-TOF identification, combining food-pathogens detection. The addition of 5 mg/kg sulfites had a limited effect, whereas increasing CO_2_ levels in the packaging significantly extended the shelf life by reducing the bacterial growth rates and prolonging the lag phases. The most affected bacteria were aerobic mesophilic and psychrotrophic bacteria, as well as *Brochothrix thermosphacta*. *Carnobacterium* spp. dominated the aerobic mesophilic group, while *Enterobacter* spp. was prevalent in *Enterobacteriaceae* and aerobic mesophilic isolates, highlighting its role in spoilage. *Hafnia alvei* was also relevant in the final spoilage stages. These results suggest the importance of these bacteria in poultry hamburger decay and demonstrate that MAP is an effective method to delay spoilage.

## 1. Introduction

Current trends in food consumption are evolving towards healthier but tasty products. New food products developed are following novel trends that focus on the production of improved formulations that are respectful of human health and environmental care, while also considering manufacturers’ interests. When talking about meat foodstuffs, there are a wide range of products at consumers’ disposal, not only unprocessed meat such as traditional steaks or whole chickens, but also processed meat products such as sausages or hamburgers, whose launch usually involves consideration of customers’ preferences [1].

Hamburgers stand out among meat products as one of the most widely accepted options among consumers of different age groups [2]. They have conventionally been formulated with bovine meat [3]. Nevertheless, reasons such as higher bovine meat production costs compared to other species, nutritional profiles of bovine meat (commonly characterized by high fat content and tough digestibility of proteins [4], or even affective preferences related to smell and taste [5] have pushed the meat industry to create new products, similar to the already existing ones, but with improvements in some of their characteristics. Indeed, a good example of improved meat products are newly formulated poultry hamburgers [6,7,8], which have entered the scene as a groundbreaking meat product that covers most of the previously cited benefits.

Thus, poultry meat has appeared as a good alternative for the formulation of hamburgers. Besides those reasons, poultry meat production has a lower environmental impact compared to other species such as bovine or pork [9]. Its nutritional profile, considering, for instance, fat and protein content, is really attractive [10,11]; it displays high digestibility [12]; and its smell and taste are more palatable, even for people who are not keen on eating meat. All these considerations make poultry meat one of the main meats consumed in Europe [13] and even worldwide [14]. Indeed, the most recent predictions published by the European Commission [13] and the Food and Agriculture Organization of the United Nations [14] estimate an increase in poultry meat consumption in the next decade.

Nevertheless, poultry meat also poses some disadvantages, as it is more perishable than meat obtained from other species. For instance, some studies describe shelf lives of 5–7 days in chicken breast preserved in cold conditions [15,16] or turkey meat [17,18], mainly due to the growth of spoilage microbiota [19], linked to modifications in their typical characteristics that make them unacceptable to consumers [20]. By contrast, longer shelf lives have been described for beef, which shows important signs of spoilage between 7 and 14 days of preservation in similar conditions [21], even minced [22]. Although both sensorial analysis and determination of chemical parameters such as pH and total volatile basic nitrogen are commonly used as indicators of meat decay [23,24], the microbiota present in meat is the basis of the deterioration [25,26]. As muscle is a sterile matrix, it can be easily inferred that microorganisms involved in poultry meat spoilage reach muscle during slaughter, skinning, evisceration, and subsequent processes [27,28]. This risk is even higher when meat products are made from minced meat, a risk exacerbated by the high concentration of nutrients and higher availability of water [29]. Hence, poultry hamburgers comply with all the requirements previously cited to boost microbial growth, and although most of the microorganisms are only implied in meat product decay, poultry hamburgers have also been involved in several reported outbreaks [30,31,32].

Regarding worrisome foodborne pathogens in hamburgers, *Salmonella* spp., *Campylobacter* spp., and *Listeria monocytgenes* are of major concern. These microorganisms mainly reach hamburgers during the production process, for instance, while mincing or mixing meat, and can cause serious illnesses. In the most recent data published by the European Food Safety Agency corresponding to 2022 [33], *Salmonella* spp. was the second causative agent, behind those unknown, while *Campylobacter* spp. was the fourth. Meanwhile, *L. monocytogenes* was the main causative agent of diseases, causing 45% of all deaths linked to foodborne outbreaks. Regarding vehicles, meat and meat products were the second-largest group involved in foodborne outbreaks, with *Salmonella* spp. and *Campylobacter* spp. being the most common causal agents. Although an accurate time–temperature cooking combination, such as 70 °C/2 min, leads to a 6-logarithm reduction in one of the most heat-resistant vegetative foodborne bacteria, *L. monocytogenes* [34], undercooking enables its survival, so the study of its presence in raw products is of extreme importance to reduce the risk of its intake.

Although sanitary issues are of major relevance, as spoilage bacteria are responsible for shortening the shelf life of animal-derived products, they are also responsible for significant economic losses, not only for meat product producers but also for consumers, as well as substantial rates of food waste [35,36,37]. Some of the most important microorganisms reported to be involved in meat decay are *Enterobacteriaceae* and *Pseudomonas* spp., the latter of which is important to consider due to its ability to grow at low temperatures and form biofilms [38], potentially eventually causing foodborne outbreaks. Additionally, there are some other microorganisms that have not traditionally attracted as much attention [39] but are currently at the top of the line, such as *Brochothrix thermosphacta* [40,41].

In order to prevent microbial growth, there are several techniques that can be applied to food products. Nevertheless, it should be taken into consideration that meat is a raw product that can suffer from strong alterations when subjected to some physical treatments such as traditional heating [42], pulsed electric fields [43,44], ultrasonication [45], or high hydrostatic pressure [46]. To avoid changes related to these processes, apart from maintenance in cold or even freezing conditions (universal recommendations for extending the shelf life of raw meat products), some other procedures are combined to increase the preservation effect, such as the modification of the packaging atmosphere or the addition of certain additives. The modification in the gas concentration contained in the package has proved to be extremely effective in postponing the spoilage of meat products, as the reduction in oxygen concentration and its exchange for inert gases is linked to a reduction in the growth rates of the aerobic populations of spoilage microbiota present in meat products [47]. Several studies have described increases in the shelf life of poultry meat products when a modified atmosphere is used in packaging. For instance, Chouliara et al. [48] found an extension of refrigerated chicken breast shelf life from 5–6 to 11–12 days.

With the aim of extending the shelf lives of meat products, one of the most common additives added are sulfites. Although they may pose specific drawbacks over consumers’ health such as hypersensitivity, allergic diseases, vitamin deficiency, and dysbiotic events of the gut and oral microbiota [49], their use in combination with other barriers such as refrigeration allows the use of a limited amount of sulfites, harmless to human health, with good results [50].

Even though there are several methods for microbial identification in foodstuffs, MALDI-TOF (Matrix-Assisted Laser Desorption/Ionization Time-of-Flight) mass spectrometry is often more fit for purpose compared to molecular techniques like PCR and sequencing due to its speed, cost-effectiveness, and ability to provide comprehensive and direct identification. It can identify microorganisms within minutes after sample preparation, while molecular techniques often require hours or days. Additionally, MALDI-TOF has lower per-sample costs since it does not require expensive reagents like primers, probes, and sequencing kits. Unlike molecular methods that require prior knowledge of target organisms and are limited by primer specificity, MALDI-TOF identifies a broad range of bacteria, yeasts, and fungi based on protein spectra, eliminating the need for DNA extraction and amplification. It is particularly effective in distinguishing closely related species, which can be challenging for molecular methods due to genetic similarities. Modern MALDI-TOF systems are equipped with extensive spectral databases that ensure high accuracy in identifying foodborne pathogens and spoilage organisms. Moreover, it aligns well with food industry needs by enabling high-throughput, real-time microbial monitoring.

Hence, the aim of this study was to evaluate the impact of modified atmosphere and addition of sulfites on the evolution of the microbiota present in poultry hamburgers and their implications on shelf life using MALDI-TOF chromatography.

## 2. Materials and Methods

### 2.1. Experimental Design

Commercial hamburgers were provided by a local enterprise. Hamburgers were formulated with chicken (64%) and turkey (12%) meat, containing 7.9% total fat, 3.9% carbohydrates, 16.6% proteins, and 1.6% salt. Hamburgers were divided into three different batches. Batch 1 was produced without the addition of sulfites and preserved with no modifications to the packaging atmosphere. In Batch 2, minced meat was supplemented with a concentration of 5 mg/kg of sulfites during the formulation and production of hamburgers, maintaining the product without any modification in the packaging atmosphere during subsequent preservation in refrigeration. Sulfites were not added to Batch 3, but the batch was packaged with a modified atmosphere of N_2_ enriched with 20% CO_2_. The packaging was made of polyethylene (PET), which is an inert material that avoids transference between the package and hamburgers, and between the packaging atmosphere and the outer environment All the hamburgers were provided frozen, and freeze-preserved at −20 °C until used.

### 2.2. Sampling and Sample Preparation

In order to study the evolution of the microbiota, hamburgers were frozen immediately after production, defrosted at day 0 and maintained in refrigeration during the study. They were sampled every 2 days from days 0 to 16. Defrosting was performed overnight while maintaining hamburgers at 5 °C. The shelf life of the poultry hamburgers was set by the manufacturer as 8 days, although an extension beyond this period was proposed for this study in order to better establish the characteristics of microbiota development. Sample preparation was performed by mixing 25 g of hamburger and 225 mL of sterile buffered peptone water at 0.1% (Oxoid LTD, Basingstoke, UK), for 5 min at 230 r.p.m. on a Stomacher 400 C (Cole-Parmer, Vernon Hills, IL, USA). After filtration, the obtained juice was collected in sterile tubes. Serial dilutions were made in sterile buffered peptone water to adjust microbial concentrations for accurate counts to be sown on Petri plates using the pour plating technique. On each sampling day, 2 hamburgers per condition were analyzed, and 2 aliquots per hamburger were studied.

### 2.3. Total Volatile Basic Nitrogen and pH

#### 2.3.1. pH

Microorganisms have diverse metabolic pathways with different by-products like lactic acid, acetic acid, succinic acid, etc., which produce acidification of the medium as well as foreign odors. Therefore, pH determination can be an early indicator of microbial alteration that alerts us that the shelf life is coming to an end [51]. To determine pH and test its usefulness as an indicator of microbial growth and end of shelf life, samples were prepared in the Stomacher as previously described and measured with a pH meter (pH-Meter BASIC 20+, Crison, Barcelona, Spain).

#### 2.3.2. Total Volatile Basic Nitrogen (TVBN)

Poultry meat is rich in proteins and peptides. Their metabolization by the present microbiota produces characteristic compounds of putrefaction, making it possible to determine the TVBN as an indicator of the microbial load and shelf life of poultry meat [51]. The volatile nitrogenous bases were extracted from the sample by means of an aqueous solution of 6% (*v*/*v*) perchloric acid. Once alkalinized, the extract was subjected to steam distillation, and the volatile basic compounds were absorbed by an acid receiver. The concentration of TVBN was determined by titration of the absorbed bases [52]. Direct distillation, a method described by Antonacopoulos [53] for the determination of TVBN in fishery products [52], was used. For TVBN determination, 2 g of each batch of hamburger sample was weighed and transferred to 50 mL of 6% perchloric acid solution. The sample was mixed using an ultrasonic technique (Ultra-Turrax T25 digital, IKAR, Staufen, Germany) until homogeneous and centrifuged at 3000 rpm at 4 °C for 5 min (Megafuge 1.0 R, Heraeus, Hanau, Germany). After centrifugation, the supernatant was collected, filtered, and transferred to a distillation tube together with 3 drops of phenolphthalein. The mixture was placed in the distiller (Kjeldahl UDK 149, Cromakit, Granada, Spain) to perform an automatic distillation with a 5% potassium hydroxide solution. Once the process ended, and the samples were recovered in a flask with 20 mL of 3% boric acid and 3 mL of Tashiro Mixed Indicator (AppliChem GmbH, Darmstadt, Germany), titration was performed with 0.1 N hydrochloric acid in an automatic pH meter (AE150, Thermo Fisher Scientific, Boston, MA, USA). As a reference, a blank trial was performed without adding the 2 g of sample.

The concentration of TVBN, expressed in mg/100 g of hamburger sample, was calculated as stated in Equation (1).

Equation (1). Assessment of TVBN per 100 g of hamburger sample.(1)$$TVBN=\frac{\left(V1-V0\right)\times 0.1\times 14\times 100}{M}$$
where

V1 = volume (mL) of 0.01 M hydrocloric acid per simple.

V0 = volume (mL) of 0.01 M hydrocloric acid per blank simple.

M = sample weight (g).

### 2.4. Bacterial Isolation

For the study of microbiota evolution, the following bacterial groups were studied as spoilage bacteria and process hygiene criteria: total aerobic mesophilic microorganisms, psychrotrophic microorganisms, *Enterobacteriaceae,* and *Brochothrix thermosphacta.* These microbial groups were investigated every 2 days between days 0 and 16, both inclusive. Additionally, *Salmonella* spp., L. *monocytogenes,* and *Campylobacter* spp. were investigated as food safety criteria. In this case, only days 0, 8, and 16 were analyzed, as these bacteria pose a food safety risk, and they should not be present in hamburgers. Also, *Pseudomonas* spp. was studied, following the same sampling pattern as a spoiling indicator.

#### 2.4.1. Bacterial Culture Media

All the culture media and selective supplements used in this study were provided by Oxoid LTD (Milan, Italy).

For the growth of total aerobic mesophilic and psychrotrophic bacteria, samples were pour-plated, and Tryptone Soy Agar supplemented with 0.6% of Yeast Extract (TSA-YE) was used as the growth medium. Streptomycin Thallous Acetate Actidione (STAA) agar, supplemented with 7,5 g/ 100 mL of glycerol and the selective supplement STAA SR0151E, was used for the selective growth of *B. thermosphacta.* Again, the samples were pour-plated. Additionally, *Pseudomonas* spp. were studied by growing them in CFC Agar, prepared by mixing *Pseudomonas* Agar Base and selective supplement SR0103.

VRBG (Violet Red Bile Glucose) Agar was used for *Enterobacteriaceae* counting. Samples were pour-plated, and after the addition of a first layer of agar, a second layer was added to obtain microaerophilic conditions. XLD Agar (Xylose Lisine Desoxycolate) was used for *Salmonella* spp. growth. The color of this medium changes when acidification occurs, and *Salmonella* colonies exhibit a characteristic red tone with a black center. L. *monocytogenes* was pour-plated in Oxoid Chromogenic *Listeria* Agar (OCLA) supplemented with selective supplements SR0226E and SR0228E. Finally, Campylobacter spp. was grown on *Campylobacter* Blood-Free Selective Agar Base (CBFSA) supplemented with selective supplement SR0155E under microaerobic conditions. Microaerobic conditions were reached in small chambers by using the Campygen Oxoid™ kit (Thermo Scientific, Loughborough, UK). Hamburger samples were in all cases pour-plated. Only in the case of *Enterobacteriaceae* was the double-layer technique used: after sowing the sample, a second layer of agar was added to create microaerophilic conditions.

#### 2.4.2. Bacterial Culture Conditions

Microbial culture conditions are presented in Table 1.

### 2.5. Identification by MALDI-TOF

MALDI-TOF^®^ Biotyper model Microflex with the software FlexControl 3.4 and MBT Compass Version 4.180 (Brüker, MA, USA) was used for bacterial identification. The reference database used was MBT Compass Explorer 4.1, which includes 10,833 reference profiles (Brüker). This equipment is intended for the characterization of bacteria by matrix-assisted laser desorption–ionization (MALDI) time-of-flight mass spectrometry (TOF/MS). Hence, this technology identifies bacteria based on the mass spectra of cells or cellular components. One of the main advantages of this method compared with traditional identification methods is its speed of analysis, as it can identify bacteria in a few minutes.

For this purpose, 215 isolates obtained from cultures on specific media and under specific conditions, derived from hamburgers with and without sulfites and with preservation in unmodified and modified atmosphere, were identified. For mesophilic aerobic bacteria, 5 colonies per plate and condition (hamburger without sulfites and with no modifications in the packaging atmosphere, hamburger with sulfites and no modifications in the packaging atmosphere, and hamburger without sulfites and supplemented with 20% CO_2_ in the packaging atmosphere) were collected on days 0, 4, 8, and 16. In order to identify the microbiota, the target days were analyzed using MALDI-TOF. *Enterobacteriaceae* were identified for days 0, 4, and 16, as their involvement in food decay is of extreme importance in the first stages of food preservation, where a fast exponential growth is commonly detected. The growth of colonies on *L. monocytogenes* and *Salmonella* spp. isolation plates was identified by MALDI-TOF for days 0, 8, and 16 in order to have a balanced overview of their potential evolution in hamburgers, as they pose a safety risk and their presence should be avoided. The same sampling was performed for the spoiling *Pseudomonas* spp. Colonies were collected directly from selective media Petri plates and were kept frozen at −20 °C in a cryoprotective solution consisting of 80% sterile peptone buffered water and 20% glycerol. All the isolates were thawed and revitalized in TSA-YE prior to MALDI-TOF identification. Preparation of bacteria isolates for MALDI-TOF identification was carried out according to the manufacturer’s guidelines. The Brüker bacterial test standard (Brüker) was used for calibration according to the manufacturer’s instructions. The identification results were interpreted according to the manufacturer’s criteria. Scores of equal or greater than 2.0 indicated species-level identification, scores of 1.700–1.999 indicated genus-level identification, and scores of <1.700 were considered not reliable. For this study, only isolates with scores equal or greater than 2.0 were considered.

### 2.6. Data Representation, Modeling, and Statistical Analysis

Bacterial evolution results were obtained from 2 replicates, 2 aliquots per replicate, and are presented as the mean value ± standard deviation. The PRISM^®^ 8.0.2 program was used for data processing and representation, as well as for statistical analysis via ANOVA (GraphPad Software, Inc., San Diego, CA, USA). Statistically significant differences were considered when *p* < 0.05.

## 3. Results and Discussion

### 3.1. TVBN and pH

From day 8, the smell of the hamburgers began to be undesirable, indicating deep decay and matching the shelf life set by the manufacturer. This day corresponds to an inflection point in pH and TVBN (Figure 1 and Figure 2). Initially, the pH decreased in the three batches and continued to decrease in the sample kept in a protective atmosphere (*p* < 0.05). On the other hand, in the samples with and without sulfites and no changes to the packaging atmosphere, it increased from day 10 and day 8, respectively, although the decrease registered in pH was more pronounced in hamburgers not subjected to any extra preservation technique (no sulfites, no protective atmosphere, (*p* < 0.05)).

In turn, the TVBN increased similarly in the three samples until day 8 (*p* > 0.05), and continued with the same trend in the sample stored in a protective atmosphere, even though differences were detected in the last phases of the study, finding higher TVBN values in hamburgers not subjected to any extra preservation technique (no sulfites, no protective atmosphere, (*p* < 0,05). In the case of the samples with and without sulfites and no changes in the packaging atmosphere, the TVBN shot up from that day onwards, matching the increase in pH.

The increase in pH described from day 8 in samples with no modifications in the packaging atmosphere may be due to protein decomposition, in parallel with the higher increase registered in TVBN, which may underlie the alkalization of the samples [24]. By contrast, the sample stored in a protective atmosphere did not experience the same increase in pH or undergo such a large increase in TVBN as the samples kept in the modified atmosphere. As stated before, unpleasant odors were considerable from day 8 onwards, matching these pH and TVBN findings. Even though the pH and TVBN behavior of hamburgers preserved in a modified atmosphere differs from the one described in the other two batches considering the whole study period (*p* < 0.05), there are no significant differences between the three batches for TVBN until day 8 (*p* > 0.05), a point that could be set as the end of the shelf life. For pH, the hamburgers followed a different trend. Even though those kept in a modified atmosphere continued to decrease in pH during the whole study period, both these and the ones supplemented with sulfites showed a similar behavior until day 8. Hence, both techniques seem to be suitable for reducing the decrease in pH (possibly associated with bacterial metabolism) during the shelf life. On the other hand, even though the modified atmosphere seems to reduce the TVBN values beyond day 8 compared to the hamburgers kept without modifications in the packaging atmosphere, there were no changes in TVBN values until day 8 (*p* > 0.05), but the rapid increase on that day could indicate the end of the shelf life. Hence, even though pH and TVBN, together with sensorial characteristics, may be used as indicators of the state of the hamburgers and may be suitable for highlighting decay, they should never be used alone, as the results were not conclusive. Bacterial counts are needed to better set the end of the shelf life and guarantee the safety of hamburgers.

### 3.2. Bacterial Counts

#### 3.2.1. Aerobic Mesophilic Bacteria

The initial total viable counts (TVCs) of aerobic mesophilic bacteria were of 5.63 ± 0.00–5.81 ± 0.12 log CFU/g at day 0, and a comparable increase in the TVCs of the three types of hamburgers led to a final TVC of 9.59 ± 0.24–10.34 ± 0.02 log CFU/g at day 16, with non-existing statistical differences between them (*p* > 0.05, Figure 3). Only lower TVCs were documented in the refrigerated hamburgers maintained in modified atmosphere conditions on days 8–14 of the shelf life of hamburgers, set by the manufacturers at 8 days. Henceforth, increasing the CO_2_ concentration of the headspace of the packaging could be an effective method to slow down bacterial development in poultry hamburgers, limiting their growth and side effects, even leading to the extension of shelf lives, even though the CO_2_ levels were not measured over the duration of this study. The increment in CO_2_ concentration in the atmosphere used for packaging prolongs the lag phase of bacterial growth and decreases the growth rate during the logarithmic phase [47,54]. For instance, a similar behavior was assessed by Patsias et al. [55], who described an extension of shelf life by more than 6 days by improving the modified atmosphere of precooked chicken products stored at 4 °C. Sulfite addition did not show any reduction in TVC in the study period.

Aerobic mesophilic bacteria comprise a diverse group of species present in meat that are able to grow under aerobic conditions. They are commonly used as indicators of food hygiene, and their determination is included in routine surveillance plans developed by meat industries [56,57]. Although the bacterial growth increase detected at the end of the study period prevented the modeling of the data, it indicated an extension of poultry hamburger shelf life when using sulfites or modified-atmosphere packaging enriched with CO_2_, with the latter approach being more fit for purpose.

#### 3.2.2. Psychrotrophic Bacteria

Aerobic psychrotrophic bacteria were not detected up to 2–4 days of maintenance in refrigeration, with the timing depending on the characteristics of the atmosphere used for packaging: whilst psychrotrophic bacterial growth began after 2 days of maintenance in refrigeration for hamburgers packaged without modifications in the atmosphere of the packaging, it started after 4 days for those packed with a modified atmosphere (Figure 4). Whereas no statistically significant differences (*p* > 0.05) were found in psychrotrophic bacterial TVCs on days 0 and 2, packaging in a modified atmosphere delayed the start of bacterial detection and resulted in lower TVCs throughout the study period until day 16 (*p* < 0.05), where slightly lower TVCs were identified compared to mesophilic bacteria (8.98 ± 0.21–9.65 ± 0.16 log CFU/g), although no statistically significant differences were detected. Addition of sulfites did not show any delay in psychrotrophic bacterial development. As was found with the TVCs of mesophilic bacteria, changes linked to modifications in the packaging atmosphere were found to have an impact on bacterial development, making this technique suitable to be considered for shelf-life extension in poultry hamburgers. The same effect has been documented for beef patties [58] or ground beef [59] preserved in refrigeration and a modified atmosphere.

Aerobic psychrotrophic bacteria comprise a group of bacteria able to grow under low-temperature conditions. They are used in routine analyses, for instance, in the dairy sector [60], vegetable production [61], or in the meat industry [29], among others, and play an important role in the monitoring of ready-to-eat products and some other refrigerated products, as small fluctuations in temperature allow significant reductions in the time to spoilage due to fast increases in psychrotrophic bacterial populations [35]. Data obtained show slight benefits in delaying and reducing the growth of psychrotrophic bacteria when modifying packaging atmosphere, which might play an important role in hamburger decay.

#### 3.2.3. *Enterobacteriaceae*

In the present study, *Enterobacteriaceae* TVCs were similar in the three conditions studied. An increase from 3.19 ± 0.3 log CFU/g at day 0 up to 9.44 ± 0.39 log CFU/g was observed on day 16, and no statistically significant differences were found between the three groups of hamburgers studied on any tested day (*p* > 0.05, Figure 5). This indicates a low efficiency of the amount of sulfites added and the modification in the gas concentration of the packaging atmosphere against this group of bacteria, facts that could be linked to a protective effect of hamburger components, such as fat or proteins [62], or to the low impact of decreasing oxygen availability on Enterobacterales, as some of the microorganisms in this group are facultative anaerobes.

*Enterobacterales* are usually used as an indicator of fecal cross-contamination and general hygiene during the production process. Although high TVCs at the beginning of the preservation period may indicate malpractices in the production process, they are commonly present in food products, and their presence does not necessarily pose a safety risk to consumers, as they are usually found in some parts of the intestinal tract of mammals, and some factors, such as the specific microorganism and the host response, play an important role in the pathogenesis [63]. Even though they pose a serious concern regarding the length of hamburgers’ shelf life, no significant improvement was observed under the conditions tested in this study.

#### 3.2.4. *Brochothrix Thermosphacta*

In this work, the initial TVCs of *B. thermosphacta* in the three groups of hamburgers were analogous (*p* > 0.05), enabling the calculation of a joint initial TVC of 3.19 ± 0.3 log CFU/g, which increased up to 8.48 ± 0.36 log CFU/g on day 16 for those hamburgers without modifications in the packaging atmosphere and 7.56 ± 0.21 log CFU/g for those with a modified atmosphere. Both values were lower than the TVCs described in the previously mentioned bacterial groups (*p* < 0.05, Figure 6). Regardless of the matching TVCs on days 0 and 16, from day 4, the TVCs of *B. thermospacta* were lower in hamburgers preserved in a modified atmosphere, reaching 1.24 log CFU/g lower TVC on day 8 (*p* < 0.05) compared with the unmodified atmosphere groups of hamburgers. This reduction was maintained throughout the entire study period. This reduction in bacterial TVC points again towards a marked effect of the packaging atmosphere composition on bacterial development. In fact, although *B. thermosphacta* is a facultative anaerobic bacterium, its growth is markedly influenced by the availability of oxygen in the packaging atmosphere [64,65]. This finding reveals that reducing oxygen in the packaging atmosphere is a good strategy to delay the decay of poultry meat hamburgers, while the addition of sulfites has less of an effect.

*B. thermosphacta* is a microorganism from the *Listeriaceae* family that is able to grow at refrigeration temperatures and is highly involved in meat spoilage, specifically in poultry meat spoilage [66,67]. It is widely spread throughout all stages of the production chain [35], and although it is facultative anaerobic, it can lead to the fastest spoilage of foodstuffs under aerobic conditions [68]. In meat products, it consumes glucose, leading to smells characterized as the smell of cheese, associated with acetoine, diacetil, and 3-metilbutanol [69]. The results obtained in this study showed the efficiency of modifying the packaging atmosphere to control the growth of *B. thermosphacta*, hence delaying hamburger decay.

#### 3.2.5. *Pseudomonas* spp.

Although bacterial growth was detected on *Pseudomonas* isolation plates, colonies did not show the typical yellowish color. Further MALDI-TOF identification was required to determine the presence of these bacteria in poultry hamburgers.

#### 3.2.6. *Salmonella* spp., *Listeria* spp., and *Campylobacter* spp.

The isolation of these three species was included in this study as an indicator of safety risk. Culture on OCLA for *Listeria* spp. pointed to the presence of two colonies of *L. monocytogenes* in one of the hamburgers at the beginning of the shelf-life study (day 0), based on their typical appearance on OCLA. Also, *Salmonella* spp. was investigated, but no colonies with the typical appearance of *Salmonella* spp. on XLD agar were found. Likewise, no typical *Campylobacter* spp. colonies were found. These data were further confirmed by MALDI-TOF.

### 3.3. Bacterial Identification

Appendix A includes all the MALDI-TOF identifications performed, and Table 2 shows a brief summary of the identification results of aerobic mesophilic bacteria included in Appendix A. All the identification results showed good identification. Regarding the spoilage microbiota corresponding to aerobic mesophilic bacteria, in hamburgers without sulfites and without modifications to their packaging atmosphere, the main identifications at day 0 corresponded to *Rothia nasimuirum* (30% of the isolates) and *Staphylococcus* spp. (30%), although *Macrococcus caseolyticus*, *Escherichia coli*, *Proteus mirabilis,* and *Corynebacterium phoceense* were also identified. This microbiota evolved to 60% *P. mirabilis* and 30% *E. coli* by day 4, 50% *Bacillus subtilis* and 50% *Carnobacterium* spp. (mainly *C. divergens*) by day 8, and increased up to 80% *Carnobacterium* spp. by day 16.

Regarding hamburgers with sulfites, 70% of the isolates corresponded to *Carnobacterium* spp. on day 0, which remained the predominant species throughout the shelf life of the hamburgers, reaching 90% of the identifications at day 8 and 100% at day 16. Other important species identified in these hamburgers were *B. subtilis* and *Staphylococcus* spp. (50% and 20% of the identifications at day 4), and *Proteus* spp. (10% at day 8). The aerobic mesophilic TVC in hamburgers without sulfites and with a modified atmosphere started with 50% *Proteus mirabilis*, 20% *Micobacterium liquefacens,* and 10% *Carnobacterium mataromaticum*, evolving towards 40% *Staphylococcus* spp., 20% *Rothia nasimurium*, 20% *Bacillus* spp., and 20% *E. coli* on day 4; 50% *Carnobacterium* spp., 20% *Bacillus* spp., and 20% *Staphylococcus* spp. on day 8; and 60% *Carnobaterium* spp. and 30% *Enterobacter* spp. on day 16.

The data obtained showed that the addition of sulfites reduced bacterial diversity, to the point where, on day 16, only *Carnobacterium* spp. was identified. This issue might be related to the sulfite reduction ability of *Carnobacterium* spp. [70], as the environmental conditions favor the growth of some specific bacteria. A similar effect was observed under the other conditions tested, as 60 and 80% of the isolates identified at day 16 in the absence of sulfites with and without modifications in the packaging atmosphere were also *Carnobacterium* spp., mainly *C. divergens*, a fact that indicates this species as one of the most relevant species concerning the spoilage of poultry hamburgers, regardless of the other protective techniques used. Predominance of *Carnobacterium* spp. during the last phases of the shelf lives of poultry meat has been previously reported [41], and it is usually considered as one of the main genera involved in meat spoilage [71]. Hence, its increase through the period evaluated and predominance in the final stages of this study further confirm *Carnobcterium* spp. as one of the main causes of spoilage. This microorganism is likely to be involved in the unexpected increase in bacterial counts detected at the end of the study period.

Likewise, *Proteus mirabilis* was highly isolated, mainly in the early days of this study. It is a common component of the normal intestinal microbiota of chicken [72], a fact that enables its transfer to the slaughter line, leading to cross-contamination, especially in evisceration processes [73]. It is an opportunistic bacterium that can cause several diseases in humans, with urinary tract infections standing out as the most prevalent [74]. Hence, it should be regarded as a concern, although it is commonly isolated from chicken meat [75,76]. Other pathogenic microorganisms, such as *Staphylococcus* spp., were quite prevalent, and although they are commonly isolated from chicken meat [77,78], they should also be kept under control, as species such as *S. epidermis* are so significant that they have even been discussed as one of the main causes of hospital-acquired bacteremia [79]. Also, *Enterobacter* spp. was relevant only in the final stages of this study in hamburgers with a modified atmosphere, a fact that could be related to its facultative anaerobic metabolism. Although it was less dominant, it should similarly be monitored as it is a frequent nosocomial infection [80].

Isolates corresponding to *Enterobacteriaceae* plates (Table 3) started with microbiota mainly composed of *Pseudomonas lundensis* (75%) and *Staphylococcus epidermis* (25%), which reveals low rates of Enterobacterales in hamburgers during the early stages of preservation after production under the tested conditions. The preservation of hamburgers without a modified atmosphere led to an increase in Enterobacterales, with *Enterobacter* spp. being the only microorganism recovered (100% of the identifications) at day 16. The addition of sulfites led to a significant increase in bacterial diversity (40% *Serratia liuqefacens*, 20% *E. coli*, 20% *Pseudomonas lundensis,* and 20% *Citrobacter freundii*) at day 4. At day 16, 40% of the bacterial isolates in hamburgers with sulfites and no modifications to the packaging atmosphere corresponded to *Hafnia alvei*, and 60% to *Enterobacter* spp. In contrast, the maintenance of hamburgers in a modified atmosphere did not have an impact as marked as the addition of sulfites on bacterial diversity, as sulfite addition reduced the identification to 80% *E. coli* and 20% *Enterobacter* spp. at day 4, with diversity further reduced at day 16, when only *Enterobacter* spp. was recovered.

As with aerobic mesophilic bacteria, *Enterobacter* spp. emerged as an important bacterium in the later stages of this study. In fact, it was the predominant species of Enterobacterales on day 16, to the extent that it was the only species detected on day 16 in hamburgers without sulfites (with *Enterobacter kobei* being the microorganism mainly detected). Other microorganisms were only detected in hamburgers with sulfites: 40% of the isolates were identified as *Hafnia alvei,* whereas 60% of the isolates were *Enterobacter* spp., specifically four different species. *Hafnia alvei* is commonly found in minced meat products, and both species are associated with the appearance of putrid off odors and/or greening of the meat [40].

Also, *Pseudomonas* spp. were identified in the early stages of maintenance. They were investigated in order to have a better view of one of the main psychrotrophic bacteria involved in the spoilage of meat-derived products (Appendix A). Data obtained showed that the main bacteria identified pertained to *Pseudomonas* spp. (50% of the isolates, corresponding to *P. aeruginosa*, *P. putida,* and *P. lundensis*), but also, *Citrobacter freundii* (44%) and *Proteus mirabilis* (6%) were identified. *Pseudomonas* spp. is a ubiquitous genus in meat products, involved in meat spoilage at cold temperatures [81,82,83]. It has been documented to cause discolorations, off odors, and slime formation [82,84], and its growth during meat storage has been associated with important sensory changes [85]. Together with *Citrobacter* spp. and *Proteus* spp., it is among the main microorganisms associated with meat spoilage [86].

Regarding bacteria used as safety risk indicators, culture on OCLA for Listeria spp. isolations revealed the presence of *L. monocytogenes* in one of the hamburgers at the beginning of the shelf-life study. This was first indicated by their typical appearance on OCLA (brilliant green colonies) and then confirmed by MALDI-TOF identification (Appendix A). Further analyses would be needed to establish the acceptability of the risk associated with hamburger consumption, as the presence of *L. monocytogenes* was linked to a unique product unit, and some other hamburgers from the same batch and/or enterprise should be analyzed in order to rule out a ubiquitous and worrisome presence of *L. moncytogenes*. Hence, although *L. monocytogenes* was identified at the initial stages of hamburger preservation, further studies would be required to consider it as a health risk and to exclude cross-contamination events during hamburger handling in bacterial microbiota studies. Other bacteria also identified in this medium were *Pseudomonas* spp. and *Rothia nasimurium*, a situation previously documented for similar selective media [87]. Both microorganisms are commonly related to meat spoilage [45].

Also, *Salmonella* spp. was investigated (Appendix A). No colonies showed the typical appearance of Salmonella spp. on XLD agar. Nevertheless, some of the isolates were identified by MALDI-TOF, finding that, although at the beginning of this study, there was marked bacterial diversity, including identification of *Staphylococcus* spp., *Pseudomonas* spp., *Kocuria* spp., or *E. coli*, at the final stages of the study, biodiversity decreased, and the main bacterium present in hamburgers without sulfites and with no modifications to their atmosphere was *Staphylococcus* spp., and *Hafnia alvei* was predominant in hamburgers with sulfites (exactly as it happened in *Enterobacteriaceae* identifications). The inclusion of modifications in the atmosphere increased bacterial diversity, identifying *Enterobacter* spp., *Carnobacterium* spp., and *Leuconostoc* spp., again indicating *Enterobacter* spp. as one of the main microorganisms involved in meat spoilage. No typical *Campylobacter* spp. colonies were found, nor were any sent for MALDI-TOF identification.

## 4. Conclusions

TVBN and pH did not show conclusive evidence of hamburger decay, supporting the need for performing comprehensive microbial studies to better understand the condition of hamburgers. The study of the spoilage microbiota revealed differences across bacterial groups and the influence of the presence of sulfites/modified packaging atmosphere. While sulfites had a minimal impact on bacterial growth, the increase in CO_2_ concentration in the packaging atmosphere led to a generalized reduction in the TVC of aerobic mesophilic and psychrotrophic bacteria, as well as *B. thermosphacta*, not only throughout the poultry hamburger shelf life but also during extended storage periods. Although the final TVCs reached were comparable, these results suggest that reducing the available oxygen in the packaging atmosphere is an effective technique to slow down bacterial growth, a fact that could form the basis for lengthening poultry shelf life and even improving the quality of poultry hamburgers. Regarding bacterial identification, *Carnobacterium* spp. was the main species detected on aerobic mesophilic bacteria isolation plates, although it should be mentioned that bacterial diversity increased in hamburgers supplemented with sulfites or maintained in a modified atmosphere. On *Enterobacteriaceae* isolation plates, the most identified bacteria were *Enterobacter*, a fact particularly evident in the last days of this study, when bacterial diversity markedly decreased. As *Enterobacter* spp. was also highly identified on aerobic mesophilic plates, it seems to play an important role in meat decay. *Hafnia alvei* was also important in the final stages of this study when sulfites were added, which could be related to a higher resistance when compared to other microorganisms. Although *L. monocytogenes* was identified at the beginning of hamburger preservation (day 0), no isolates were detected in the subsequent days, implying irrelevance concerning food safety. All these findings suggest the involvement and importance of bacteria such as *Carnobacterium* spp., *Enterobacter* spp., or even *Hafnia alvei* in spoilage, and the suitability of atmosphere modification for the control and postponement of poultry hamburger decay.

## Figures and Tables

**Figure 1 microorganisms-13-00754-f001:**
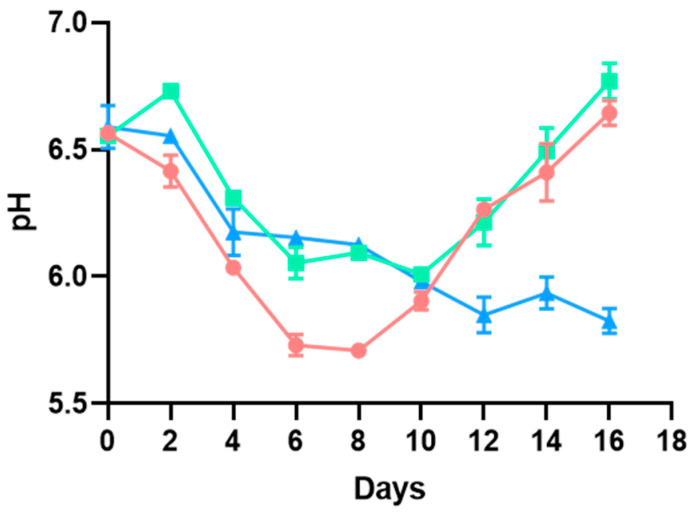
Evolution of the pH described for poultry hamburgers without sulfites and with no changes in the packaging atmosphere (●), with sulfite addition and no changes in the packaging atmosphere (◼), and without sulfites but with a modified atmosphere (▲). *p* < 0.05.

**Figure 2 microorganisms-13-00754-f002:**
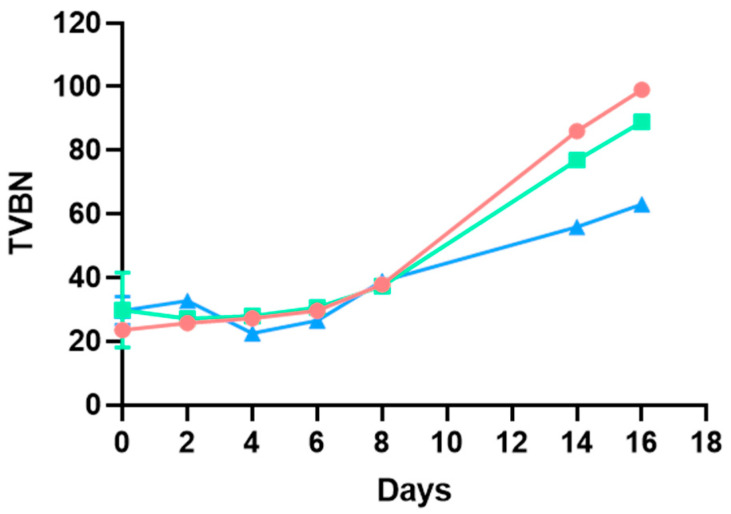
Evolution of the TVBN described for poultry hamburgers without sulfites and with no changes in the packaging atmosphere (●), with sulfite addition and no changes in the packaging atmosphere (◼), and without sulfites but with a modified atmosphere (▲). *p* < 0.05.

**Figure 3 microorganisms-13-00754-f003:**
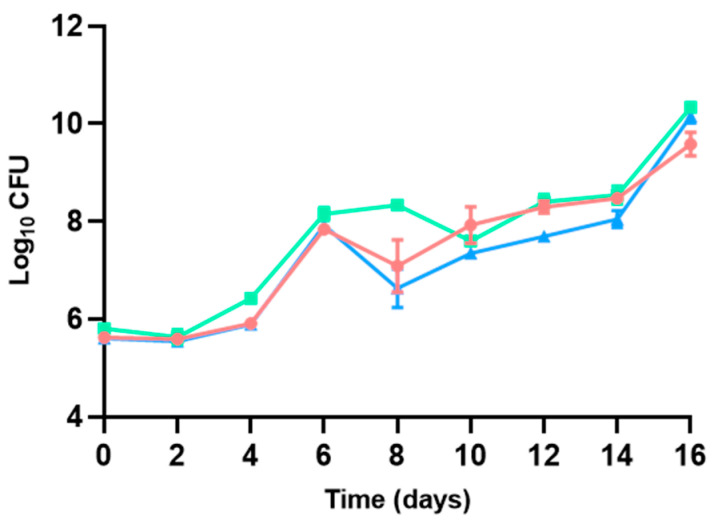
Aerobic mesophilic TVC described for poultry hamburgers without sulfites and with no changes in the packaging atmosphere (●), with sulfite addition and no changes in the packaging atmosphere (◼), and without sulfites but with a modified atmosphere (▲).

**Figure 4 microorganisms-13-00754-f004:**
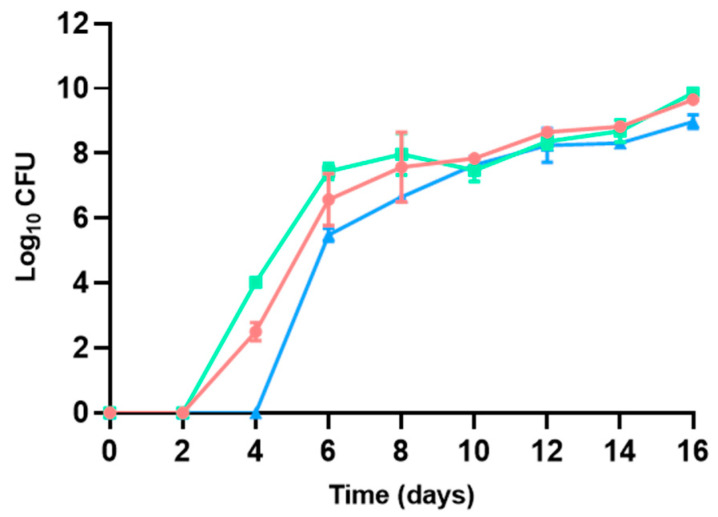
Aerobic psychrotrophic TVC described for poultry hamburgers without sulfites and with no changes in the packaging atmosphere (●), with sulfite addition and no changes in the packaging atmosphere (◼), and without sulfites but with a modified atmosphere (▲). *p* < 0.05.

**Figure 5 microorganisms-13-00754-f005:**
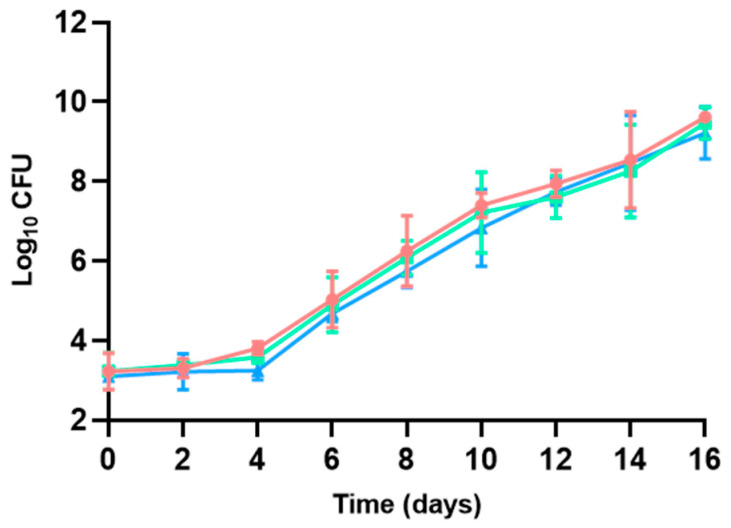
*Enterobacteriaceae* TVC described for poultry hamburgers without sulfites and with no changes in the packaging atmosphere (●), with sulfite addition and no changes in the packaging atmosphere (◼), and without sulfites but with a modified atmosphere (▲).

**Figure 6 microorganisms-13-00754-f006:**
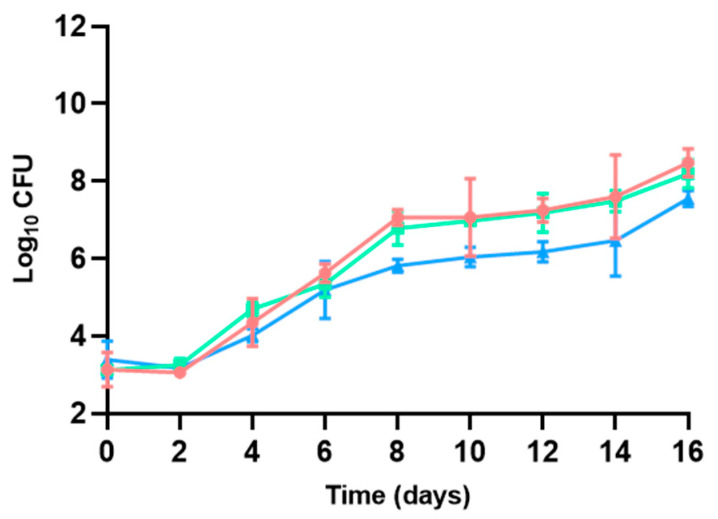
*B. thermosphacta* TVC described for poultry hamburgers without sulfites and with no changes in the packaging atmosphere (●), with sulfite addition and no changes in the packaging atmosphere (◼), and without sulfites but with a modified atmosphere (▲). *p* < 0.05.

**Table 1 microorganisms-13-00754-t001:** Culture media and incubation conditions used for each microbial group, matching the requirements of the ISO standards.

Microbial Group	Medium	Temperature (°C)	Incubation Time (Days)	Atmosphere	ISO Standards
Mesophilic	TSA-YE	35	1	Aerobiosis	UNE-EN ISO 4833
Psychrotrophic	TSA-YE	7	7	Aerobiosis	UNE-EN ISO 4833
*Enterobacteriaceae*	VRBG	35	1	Aerobiosis	UNE-EN ISO 21528
*B.thermosphacta*	STAA	25	2	Aerobiosis	-
*Pseudomonas* spp.	CFC	20	2	Aerobiosis	-
*Salmonella* spp.	XLD	35	1	Aerobiosis	UNE-EN ISO 6579
*L. monocytogenes*	OCLA	10	1	Aerobiosis	UNE-EN ISO 11290
*Campylobacter* spp.	CBFSA	40	1	Microaerophilia	EN-ISO 10.272-2

**Table 2 microorganisms-13-00754-t002:** Summary of the identification results of aerobic mesophilic bacteria included in Appendix A and obtained by MALDI-TOF.

Sulfites	Atmosphere	Day 0		Day 4		Day 8		Day 16	
		% Isolates	Identification	% Isolates	Identification	% Isolates	Identification	% Isolates	Identification
W/O *	Unmodified	30	*Rothia nasimurium*	60	*Proteus mirabilis*	50	*Carnobacterium* spp.	80	*Carnobacterium* spp.
		30	*Staphylococcus* spp.	30	*E.coli*	50	*B. subtilis*	10	*Leuconostoc mesenteroides*
		10	*Macrococcus caseolyticus*	10	*Staphylococcus simulans*			10	*Kurthia zopfii*
		10	*Escherichia coli*						
		10	*Proteus mirabilis*						
		10	*Corynebacterium phoceense*						
W *	Unmodified	70	Carnobacterium spp	50	*B. subtilis*	90	*Carnobacterium divergens*	100	*Carnobaterium* spp.
		20	*Staphylococcus* spp.	20	*Staphylococcus simulans*	10	*Proteus mirabilis*		
		10	*Enterococcus faecalis*	10	*Enterococcus faecalis*	10	*Proteus mirabilis*		
				10	*Pseudomonas lundensis*				
				10	*Carnobacterium divergens*				
W/O *	Modified	50	*Proteus mirabilis*	40	*Staphylococcus* spp.	50	*Carnobacterium* spp.	60	*Carnobacterium divergens*
		20	*Microbacterium liquefacens*	20	*Rothia nasimuirum*	20	*Bacillus* spp.	30	*Enterobacter* spp.
		10	*Carnobacterium maltaromaticum*	20	*Bacillus* spp.	20	*Staphylococcus* spp.	10	*Leuconostoc mesenteroides*
		10	*Rothia nasimurium*	20	*E. coli*	10	*E.coli*		
		10	*Escherichia coli*						

* W/O (without): hamburgers without sulfites. W (with): hamburgers with 5 mg/kg of sulfites.

**Table 3 microorganisms-13-00754-t003:** Summary of the identification results of *Enterobacteriaceae* included in Appendix A, obtained by MALDI-TOF.

Sulfites	Atmosphere	Day 0		Day 4		Day 16	
		% Isolates	Identification	% Isolates	Identification	% Isolates	Identification
W/O *	Unmodified	75	*Pseudomonas lundensis*			100	*Enterobacter* spp.
		25	*Staphylococcus epidermis*				
W *	Unmodified	75	*Pseudomonas lundensis*	40	*Serratia liquefaciens*	60	*Enterobacter* spp.
		25	*Staphylococcus epidermis*	20	*Pseudomonas lundensis*	40	*Hafnia alvei*
				20	*Citrobacter freundii*		
				20	*Escherichia coli*		
W/O *	Modified	75	*Pseudomonas lundensis*	80	*Escherichia coli*	100	*Enterobacter* spp.
		25	*Staphylococcus epidermis*	20	*Hafnia alvei*		

* W/O (without): hamburgers without sulfites. W (with): hamburgers with 5 mg/kg of sulfites.

## Data Availability

The data that support the findings of this study are available on request.

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
