# Peer review of "Growth and Diversity of Spoiling and Foodborne Bacteria in Poultry Hamburgers in Modified Atmosphere and with Sulfites During Shelf Life"

_microorganisms, 2025, doi:10.3390/microorganisms13040754_

Round 1
Reviewer 1 Report
Comments and Suggestions for Authors
Thank you for having an opportunity to review manuscript entitled "Influence of Sulphites and Modified Atmosphere Over the Growth and Diversity of Spoiling and Foodborne Bacteria in Poultry Hamburgers"
The main objective of this study was to evaluate and compare the effects of adding sulfites and using a modified atmosphere on the microbial quality and safety of refrigerated fresh poultry hamburgers.
The manuscript's language and purpose are clear and comprehensible, rendering it accessible to a broader audience. The materials and methods section are thorough and offer full information regarding the experimental design, bacterial isolation, and identification procedures. Also, authors employed pertinent statistical analysis (ANOVA) and suitable tools, so ensuring the reliability of the data analysis is unquestionable. The results are clearly presented in tables and figures, and the discussion offers context and interpretation of the findings.
However, main issue of this manuscript, in my humble opinion, is lack of novelty. The study corroborates earlier findings about the efficacy of modified atmospheric packaging (MAP) and sulphites but does not present substantial new insights or methodology. The application of MAP, especially with increased CO2 concentrations, is an established technique in food preservation for decades. Numerous studies have already established its efficacy in prolonging the shelf life of meat products, especially poultry. Although the study may have investigated certain CO2 concentrations or packing materials, the overarching notion is not innovative.
Next, sulfites have historically served as preservatives in meat products. Examining their influence on poultry hamburgers, although pertinent to this particular product, does not constitute an innovative methodology. Finally, evaluating alterations in microbial numbers and identifying predominant spoilage bacteria are typical practices in food science. The specific microbial species may differ based on the product and environment, although the fundamental process is consistently employed.
What I also find missing in the manuscript is lack of exploration of quality parameters like sensory attributes and chemical changes, since these are interconnected with the changes in bacterial population in the sense shelf life of this meat product. Solely focusing to bacteria does not contribute to the novelty.
This manuscript would be improved by further addressing the aforementioned flaws to augment its effect and contribution to the field.
Author Response
Thank you for having an opportunity to review manuscript entitled "Influence of Sulphites and Modified Atmosphere Over the Growth and Diversity of Spoiling and Foodborne Bacteria in Poultry Hamburgers"
The main objective of this study was to evaluate and compare the effects of adding sulfites and using a modified atmosphere on the microbial quality and safety of refrigerated fresh poultry hamburgers.
The manuscript's language and purpose are clear and comprehensible, rendering it accessible to a broader audience. The materials and methods section are thorough and offer full information regarding the experimental design, bacterial isolation, and identification procedures. Also, authors employed pertinent statistical analysis (ANOVA) and suitable tools, so ensuring the reliability of the data analysis is unquestionable. The results are clearly presented in tables and figures, and the discussion offers context and interpretation of the findings.
However, main issue of this manuscript, in my humble opinion, is lack of novelty. The study corroborates earlier findings about the efficacy of modified atmospheric packaging (MAP) and sulphites but does not present substantial new insights or methodology. The application of MAP, especially with increased CO2 concentrations, is an established technique in food preservation for decades. Numerous studies have already established its efficacy in prolonging the shelf life of meat products, especially poultry. Although the study may have investigated certain CO2 concentrations or packing materials, the overarching notion is not innovative.
Next, sulfites have historically served as preservatives in meat products. Examining their influence on poultry hamburgers, although pertinent to this particular product, does not constitute an innovative methodology. Finally, evaluating alterations in microbial numbers and identifying predominant spoilage bacteria are typical practices in food science. The specific microbial species may differ based on the product and environment, although the fundamental process is consistently employed.
Dear reviewer, first of all, thanks for the time invested in the article reading, your evaluation and interpretation of the results and conclusions obtained from the study content Of course we are aware that MAP and sulphites are not novel ways of increasing the shelf life of meat products maintained in refrigeration. Both techniques have been proved to be effective when enlarging shelf-lives of a great variety of meat products, and the study did not focus on proving their ability as preserving technology, but to see the implications of their use over the spoiling and eventual foodborne microbiota present in the poultry hamburgers studied. Even though we did not discover any new food preservation method, we used groundbreaking methods to study the influence of effective and consolidated techniques commonly employed in the extension of meat products shelf-lives in the microbiota and their evolution, with can set the basis for further strategies targeted in certain microbiota more implicated in meat decay. We understand your view of the article and we thank you for having extracted this assessment, but as the focus was not proving the effect of these techniques, we please ask you to reconsider your evaluation under this other prism.
Regarding the issue of bacteria found and bacterial counts, it is important to bear in mind that meat is a sterile product. Hence, all the microbiota found on it is coming from external sources that may thoroughly vary between meat products depending on several factors such as the kind of preparation, ingredients, manufacturers, kind of meat industry, etcetera. Nevertheless, in this specific product, we see the evolution of the initial microbiota and counts and the influence on them of both preservation techniques. Like it should be done in all the studies published, we perfectly describe the product where the study was performed, from where it is easily exerted the limitations it has when results are transferred to a different product. Nevertheless, the modifications on bacterial diversity and microbiota found in the study are aligned with previous studies (Raimondi et al., 2019; Duthoo et al., 2022), hence supported by other research performed in the field. The authors thank you again for your critical analysis of the study and aim you to reconsider your view aligning with our vision.
- Raimondi, S., Luciani, R., Sirangelo, T. M., Amaretti, A., Leonardi, A., Ulrici, A., ... & Rossi, M. (2019). Microbiota of sliced cooked ham packaged in modified atmosphere throughout the shelf life: Microbiota of sliced cooked ham in MAP. International Journal of Food Microbiology, 289, 200-208.
- Duthoo, E., De Reu, K., Leroy, F., Weckx, S., Heyndrickx, M., & Rasschaert, G. (2022). To culture or not to culture: careful assessment of metabarcoding data is necessary when evaluating the microbiota of a modified-atmosphere-packaged vegetarian meat alternative throughout its shelf-life period. Bmc Microbiology, 22(1), 34.
What I also find missing in the manuscript is lack of exploration of quality parameters like sensory attributes and chemical changes, since these are interconnected with the changes in bacterial population in the sense shelf life of this meat product. Solely focusing to bacteria does not contribute to the novelty.
Dear reviewer, indeed, we performed several analysis regarding pH changes and NBVT we did not include in the paper in order not to make it too cumbersome for the readers, as the main objective was to study the changes in the microbiota. We have the available on request, and so we have pointed out on lines 132-134. I also send the results to you attached on a file. Regarding the point of sensory attributes, we did not count with the corresponding authorization required by our organization to perform any tasting, so it is more a bureaucratic issue we could not solve.
Evolution of the pH described for poultry hamburgers without sulphites and no changes on the packaging atmosphere (l), with sulphite addition and no changes on the packaging atmosphere (n) and without sulphites but with modified atmosphere (â–²).
Evolution of the TVB-N described for poultry hamburgers without sulphites and no changes on the packaging atmosphere (l), with sulphite addition and no changes on the packaging atmosphere (n) and without sulphites but with modified atmosphere (â–²).
This manuscript would be improved by further addressing the aforementioned flaws to augment its effect and contribution to the field.
Dear reviewer, thanks again for your valuable time and assessment of the study. It has helped us to improve the content and re-consider several aspects of the content included. Moreover, it will help us to improve and try to extend the study in further research.

Reviewer 2 Report
Comments and Suggestions for Authors
This article is highly relevant for the food industry, but trivial for mass spectrometric analytics. Therefore, the authors did the right thing by submitting it to a microbiological journal.
MALDI biotyper really helps to distinguish between these microorganisms. However, both growth and lag phases can also be used for this (as can be seen from the graphs).
All methods are used quite correctly.
The work does not have any fundamental novelty, but it has a clearly expressed applied value.
Therefore, it can be published in its current form.
Author Response
The authours sincerely thank you for your assessment and words of our work. Under our view, it gives a thoroughly overview of meat microbiota diveristy through the shelf life of the product studied and may be useful for other researchers in the fiel as a basis for the devlopment of their studies. Thanks again for your time an valuable opinion.
Reviewer 3 Report
Comments and Suggestions for Authors
The study "Influence of sulphites and modified atmosphere over the growth and diversity of spoiling and foodborne bacteria in poultry hamburgers" was written by González-Fandos et al.
The authors of this study assessed both strategies by using MALDI-TOF identification and total viable counts to analyze the bacterial development in chicken hamburgers. While adding 5 mg/kg sulphites had no effect, raising the amount of CO2 in packaging greatly increased shelf life by slowing down bacterial development and extending lag periods. The authors claim that Brochothrix thermosphacta and aerobic mesophilic and psychrotrophic bacteria were the most impacted. While Enterobacter spp. was common in Enterobacteriaceae and aerobic mesophilic isolates, emphasizing its involvement in spoiling, Carnobacterium spp. dominated the aerobic mesophilic group.
- Lines 149-150. Mesophilic and psychrophilic bacteria included exactly which taxa?
- Why were chosen exactly Enterobacteriaceae, thermosphacta, Pseudomonas spp., Salmonella spp., L. monocytogenes and Campylobacter spp.?
- There should be limitations of this research, since the most of microorganisms are uncultivated and authors do not know about whole composition of microbiome of hamburgers.
- MALDI-TOF identification is good, but it did not allow detect some bacteria exactly on the species level.
- It would be better if the authors provided an illustration of what the microbiota of hamburgers would be like in ideal and non-ideal storage conditions.
- Line 368. Are Carnobacterium sulphite consumers? Why they are resistant to sulphite? Are there anything?
- Table 3. How do authors explain that in the 0 days, Hafnia alvei were not found, and then they are appeared?
- Table 2. There is more diversity of bacteria in 8 days than in 0 days. Please explain this.
Author Response
The study "Influence of sulphites and modified atmosphere over the growth and diversity of spoiling and foodborne bacteria in poultry hamburgers" was written by González-Fandos et al.
The authors of this study assessed both strategies by using MALDI-TOF identification and total viable counts to analyze the bacterial development in chicken hamburgers. While adding 5 mg/kg sulphites had no effect, raising the amount of CO2 in packaging greatly increased shelf life by slowing down bacterial development and extending lag periods. The authors claim that Brochothrix thermosphacta and aerobic mesophilic and psychrotrophic bacteria were the most impacted. While Enterobacter spp. was common in Enterobacteriaceae and aerobic mesophilic isolates, emphasizing its involvement in spoiling, Carnobacterium spp. dominated the aerobic mesophilic group.
- Lines 149-150. Mesophilic and psychrophilic bacteria included exactly which taxa?
For TVC in mesophilic and psychrophilic bacteria, no specific microorganisms were explored, only the total number of microorganisms able to grow in the conditions tested as an exploration of the hygienic state of the hamburgers.
- Why were chosen exactly Enterobacteriaceae, thermosphacta, Pseudomonas spp., Salmonella spp., L. monocytogenes and Campylobacter spp.?
Enetrobacteriaceae was chosen as a common indicator of faecal contamination and hygiene parameter in meat. B. thermosphacta was chosen because of there are a lot of recent studies that point it out as a key bacterium in meat spoilage, and we considered of great relevance incluiding its exploration in these newly formulated hamburgers. On their hand, Salmonella spp., L. monocytogenes and Campylobacter spp. were included as they are the most worrisome microorganisms in foodborne outbreaks, and they are foodborne agents commonly found in poultry meat, even more in those made from minced meat.
- There should be limitations of this research, since the most of microorganisms are uncultivated and authors do not know about whole composition of microbiome of hamburgers.
The approach of the study was not finding the whole microbiome of the hamburgers by using whole genome sequencing or other genetic techniques. We wanted to see the effect of several ways of maintenance after production and packaging and the evolution of some specific and relevant microbiota, so that it was a targeted study where we wanted to see the evolution of some microbiota using the techniques we had at our disposal. We have used this technique in several previous studies with good results:
- da Silva-Guedes, J., Martinez-Laorden, A., Gonzalez-Fandos, E. (2022). Effect of the presence of antibiotic residues on the microbiological quality and antimicrobial resistance in fresh goat meat. Foods, 11,3030.
- Martinez-Laorden, A., Arraíz-Fernandez, C. Gonzalez-Fandos, E. (2023). Microbiological quality and safety of fresh turkey meat at retail level including the presence of ESBL-producing Enterobacteriaceae and methicillin-resistant aureus. Foods 12, 1274. https://doi.org/10.3390/foods12061274
- Martinez-Laorden, A., Arraíz-Fernandez, C. Gonzalez-Fandos, E. (2023). Microbiological quality and safety of fresh quail meat at the retail level Microorganisms 11, 2213. https://doi.org/10.3390/microorganisms11092213.
- Da Silva-Guedes, J., Velilla-Rodrigues, Gonzalez-Fandos, E. (2024). Microbiological Quality and Safety of Fresh Rabbit Meat with Special Reference to Methicillin-Resistant aureus (MRSA) and ESBL-Producing E. coli. Antibiotics 2024, 13, 256. https://doi.org/10.3390/antibiotics13030256
- MALDI-TOF identification is good, but it did not allow detect some bacteria exactly on the species level.
We had a great rate of identification by the methodology used, and in the cases we were not able to determine the species, this result din not pose any disadvantage to our study, as most of the families involved in meat spoilage spoil in a similar way and exert similar effect at a family level. We have an extent experience in using MALDI-TOF identification at species level in meat with good results as can be seen in previous studies carried out
- Martinez-Laorden, A., Arraíz-Fernandez, C. Gonzalez-Fandos, E. (2023). Microbiological quality and safety of fresh turkey meat at retail level including the presence of ESBL-producing Enterobacteriaceae and methicillin-resistant S. aureus. Foods 12, 1274. https://doi.org/10.3390/foods12061274
- Martinez-Laorden, A., Arraíz-Fernandez, C. Gonzalez-Fandos, E. (2023). Microbiological quality and safety of fresh quail meat at the retail level Microorganisms 11, 2213. https://doi.org/10.3390/microorganisms11092213.
- It would be better if the authors provided an illustration of what the microbiota of hamburgers would be like in ideal and non-ideal storage conditions.
Thank you for the suggestion. Regarding this issue, it is important to bear in mind that meat is a sterile product. Hence, all the microbiota found on it is coming from external sources that may thoroughly vary between meat products depending on several factors such as the kind of preparation, ingredients, manufacturers, kind of meat industry, etcetera. Indeed, we have done a reference to this subject on lines 63-66, but we cannot adventure more.
- Line 368. Are Carnobacterium sulphite consumers? Why they are resistant to sulphite? Are there anything?
Thank you for your comment. We have added a little explanation aout the issue in line 370.
- Trutko, S. M., Dorofeeva, L. V., Shcherbakova, V. A., Chuvil’skaya, N. A., Laurinavichus, K. S., Binyukov, V. I., ... & Akimenko, V. K. (2005). Occurrence of nonmevalonate and mevalonate pathways for isoprenoid biosynthesis in bacteria of different taxonomic groups. Microbiology, 74, 153-158.
- Holzapfel, W. H. (1998). The Gram-positive bacteria associated with meat and meat products.The microbiology of meat and poultry, 31, 35-74.
- Schillinger, U., & Holzapfel, W. H. (1995). The genus Carnobacterium. In The genera of lactic acid bacteria (pp. 307-326). Boston, MA: Springer US.
- Table 3. How do authors explain that in the 0 days, Hafnia alvei were not found, and then they are appeared? Thank you for the comment.
We already realized this issue, but as the common saying in Microbiology that points out that “zero sterility” does not exist, we assume that the initial bacterial counts of some of the bacteria were so low that they passed undetected at the first stages of the shelf-life, even more when no pre-enrichment or specific targeted methods were not carried out as this microorganism was not specifically screened.
- Table 2. There is more diversity of bacteria in 8 days than in 0 days. Please explain this.
We consider that this point is extremely related to the previous one. Even though the diversity increases as the shelf-life goes on, more than the diversity itself is the diversity of microbiota detected. As microorganisms have more time to develop, it is much easier to detect them. When bacterial counts are extremely lower, the ability to detect bacteria with low bacterial counts decreases, and due to this effect, bacterial diversity detected increases, but it might decrease when conditions favor the growth of some specific bacteria. We have included an explanation about this issue in Line 371-372. Thanks again for your valuable time dedicated to read and evaluate the study, and all your worthwhile comments and suggestions.

Round 2
Reviewer 1 Report
Comments and Suggestions for Authors
Dear author,
Thank you very much for your feedback and efforts done to discuss issues I have raised pertaining to your manuscript.
Indeed, I have understood very well what your objective during the initial review was (i.e. microbiota during the shelf-life), and reassesed your arguments, however, manuscript written "as is" leads to the aforementioned concerns.
Now, as stated in my review, there exists the potential for your paper to make a significant scientific contribution to the subject. I am willing to reevaluate my perspectives if the manuscript undergoes significant revision and is reorganized to better line with MDPI's objectives.
The title could be rephrased to commence with, for example, "Microbiota of Poultry Hamburgers in MAP and Sulphites During Shelf-Life" or a comparable variation.
Lines 85-93 necessitate expansion to more comprehensively characterize possible microflora. The current length is insufficient, and your primary purpose was to do a study on this matter. The remainder of the Introduction section is satisfactory.
In the Materials and Methods section, additional parameters about shelf life must be incorporated, as they are referenced multiple times throughout the text. Therefore, pH testing and TVB-N methodologies must be clearly specified and thoroughly elucidated. Furthermore, it is imperative to scientifically address the absence of organoleptic testing. In its absence, how could readers comprehend the standard shelf life in Batch 1, which functioned as a control? Merely depending on the manufacturers' 8-10 day timeframe lacks scientific rigor, as you even conjectured in Lines 139-140.
The primary concern is the absence of data regarding CO2/N2 concentration measurements conducted by Oxybaby or comparable devices. The necessity of this is linked to your findings; for instance, Line 230 addresses and discusses CO2 increase, although, in the existing edition, readers are unaware of the specific concentration. Particularly, as CO2 dissolves in the aqueous phase of meat, its initial concentration diminishes. Kindly reevaluate this, as it does impact microbiota.
Section 2.3.2 necessitates additional information regarding Bruker's databases and versions to enhance the transparency and reproducibility of your experiment.
Growth curve figures exhibiting p<0.05 (e.g., psychrotrophic total viable count) should be annotated with asterisks or letters for enhanced clarity.
MALDITOF tables are data-rich although rather perplexing due to the abundance of information; I believe the text would be more comprehensible if microbiota changes were shown through supplementary figures, such as pie charts or similar visualizations.
Finally, a comprehensive discussion and comparison of microbiome data acquired through molecular methods (rep-PCR, fingerprinting, NGS) versus MALDI-TOF would provide a conclusive perspective, as MALDI-TOF is significantly less labor-intensive than NGS or comparable techniques. The objective would be to scientifically evaluate and advocate for MALDI-TOF as a more suitable instrument for food microbiologists.
Author Response
Dear author,
Thank you very much for your feedback and efforts done to discuss issues I have raised pertaining to your manuscript.
Indeed, I have understood very well what your objective during the initial review was (i.e. microbiota during the shelf-life), and reassesed your arguments, however, manuscript written "as is" leads to the aforementioned concerns.
Now, as stated in my review, there exists the potential for your paper to make a significant scientific contribution to the subject. I am willing to reevaluate my perspectives if the manuscript undergoes significant revision and is reorganized to better line with MDPI's objectives.
The title could be rephrased to commence with, for example, "Microbiota of Poultry Hamburgers in MAP and Sulphites During Shelf-Life" or a comparable variation.
Thank you for your suggestion. We did not study the whole microbiota using WGS or similar techniques, but performed targeted analysis of the microbiota for the evaluation of the most important microbial aspects involved in meat products decay and safety. Hence, we have changed the title as it is currently set to better fit your inquiries and our purpose.
Lines 85-93 necessitate expansion to more comprehensively characterize possible microflora. The current length is insufficient, and your primary purpose was to do a study on this matter.
The remainder of the Introduction section is satisfactory. In the previous lines we deal with safety concern microorganisms. Authors consider extending the dissertation about microbiota in the introduction would be unnecessary considering the length of the discussion section. Please, authors ask you to take this consideration and contemplate the paper as a whole. This way the paper comprehensively deals with this issue.
In the Materials and Methods section, additional parameters about shelf life must be incorporated, as they are referenced multiple times throughout the text. Therefore, pH testing and TVB-N methodologies must be clearly specified and thoroughly elucidated.
Dear reviewer, I guess you are asking us to include the methodology and results of pH and TVB-N in the paper. So we have done, looking for its upgrading. Thank you for aiming us to improve it.
Furthermore, it is imperative to scientifically address the absence of organoleptic testing. In its absence, how could readers comprehend the standard shelf life in Batch 1, which functioned as a control? Merely depending on the manufacturers' 8-10 day timeframe lacks scientific rigor, as you even conjectured in Lines 139-140.
Dear reviewer, we have answered your inquiries in the introduction. Regarding the shelf-life, we have to bear in mind that this poultry hamburgers were provided by a local manufacturer that gave us that information. Private enterprises are sometimes quite cautious with the data they share with scientists. Moreover, organoleptic testing was not contemplated at any point of the study, as it was not the point of the article. We wanted to study the microbiota, not determining the shelf life. Hence we did not consider necessary the inclusion of organoleptic testing and so we did.
The primary concern is the absence of data regarding CO2/N2 concentration measurements conducted by Oxybaby or comparable devices. The necessity of this is linked to your findings; for instance, Line 230 addresses and discusses CO2 increase, although, in the existing edition, readers are unaware of the specific concentration. Particularly, as CO2 dissolves in the aqueous phase of meat, its initial concentration diminishes. Kindly reevaluate this, as it does impact microbiota.
Dear reviewer, thank you for your considerations. At the current moment we are not in terms of doing the measures as we consider that some other factors might influence the data obtained and the correlations with already existing results would not be accurate enough. Moreover, data provided are extensive enough to provide information about the issue proposed in the study. Hence, even though this factors might influence the diversity and growth of the microbiota, the goal of the study was not evaluating the influence of the factors derived from the preservation technique over the microbiota, but to study the microbiota itself. In order to better understand this point, we have changed the title of the paper as you suggested in your first comment, and to fix your suggestions, we have added the available TVBN and pH data.
Section 2.3.2 necessitates additional information regarding Bruker's databases and versions to enhance the transparency and reproducibility of your experiment.
Thank you for your suggestion. We have fixed it. The description has been included in lines 255-258.
Growth curve figures exhibiting p<0.05 (e.g., psychrotrophic total viable count) should be annotated with asterisks or letters for enhanced clarity.
Thank you for your suggestion. We have fixed it.
MALDITOF tables are data-rich although rather perplexing due to the abundance of information; I believe the text would be more comprehensible if microbiota changes were shown through supplementary figures, such as pie charts or similar visualizations.
MALDITOF tables content is supported by the text. We considered including more figures but the diversity of the microbiota did not allow a good interpretation of results, so after reconsidering several times the best way to summarize the information, we opted by doing it this way. Nevertheless, thank you for the deep reading and evaluation of the possibilities of data presentation that better suit to the understanding of the paper.
Finally, a comprehensive discussion and comparison of microbiome data acquired through molecular methods (rep-PCR, fingerprinting, NGS) versus MALDI-TOF would provide a conclusive perspective, as MALDI-TOF is significantly less labor-intensive than NGS or comparable techniques.
The objective would be to scientifically evaluate and advocate for MALDI-TOF as a more suitable instrument for food microbiologists. Thank you for your suggestion. We have included a mention of the issue on the instruction, from lines 125 to 140.

Reviewer 3 Report
Comments and Suggestions for Authors
All my comments were adressed.
Author Response
The authors thank you for your time and efforts leading to the improvement of the study.
Round 3
Reviewer 1 Report
Comments and Suggestions for Authors
Dear Authors,
Thank you for putting efforts to address issues and concerns raised in previous review version. You have done considerable changes in the manuscript, and now it reads much better.
As I indicated in the previous review, we, the reviewers, fully understand what your aim in this manuscript is. And if you go through the review, you will see that the "core" microbiota results and discussion pertaining to your primary aim (and new insights) are actually positively evaluated, given the plethora of growth and MALDITOF data produced.
In evaluation of overall strengths and weaknesses of the scientific merit and research approach in your manuscript, issues pertaining to the factors that might influence microbiota dynamics in hamburgers still remains, since, as you mentioned, a commercial company produced hamburgers, packed them in unknown MAP and added certain amounts of sulphites, provided you with limited production data and so on, and you, as a researchers, didn't have a full analytical control. Since some of these factors affects outcomes of your primary aim (as you speculate e.g. effect of CO2 in growth of microbiota in Line 348), and you are effectively not having data or control over them, this neccessitates short explanation for the scientific audience in e.g. simple form of "limitations of the study" statement or similar in order not to make your extensive microbiome data to appear disconnected from the context of microbiome <--> shelf-life changes complex interaction.
Next, I strongly hold my view and advise authors to add a short section on the spoilage bacteria in meat preparations in the Introduction. Your Discussion section is quite all right, but some background and current overview on spoilage bacteia need to be presented in Introduction.
As you might understand, the goal of our review comments and suggestions is to reduce weakness (by addressing obvious limitations) in the manuscript, but also to support your main findings. So, no need to repeatedly advocate that aim are microbiota and not the shelf-life, we see that, and we hope you would accept these benevolent suggestions.
Author Response
Dear Authors,
Thank you for putting efforts to address issues and concerns raised in previous review version. You have done considerable changes in the manuscript, and now it reads much better.
As I indicated in the previous review, we, the reviewers, fully understand what your aim in this manuscript is. And if you go through the review, you will see that the "core" microbiota results and discussion pertaining to your primary aim (and new insights) are actually positively evaluated, given the plethora of growth and MALDITOF data produced.
Thank you very much for your words and support.
In evaluation of overall strengths and weaknesses of the scientific merit and research approach in your manuscript, issues pertaining to the factors that might influence microbiota dynamics in hamburgers still remains, since, as you mentioned, a commercial company produced hamburgers, packed them in unknown MAP and added certain amounts of sulphites, provided you with limited production data and so on, and you, as a researchers, didn't have a full analytical control. Since some of these factors affects outcomes of your primary aim (as you speculate e.g. effect of CO2 in growth of microbiota in Line 348), and you are effectively not having data or control over them, this neccessitates short explanation for the scientific audience in e.g. simple form of "limitations of the study" statement or similar in order not to make your extensive microbiome data to appear disconnected from the context of microbiome <--> shelf-life changes complex interaction.
Thank you for your suggestion. We have included that mention in lines 350-351.
Next, I strongly hold my view and advise authors to add a short section on the spoilage bacteria in meat preparations in the Introduction. Your Discussion section is quite all right, but some background and current overview on spoilage bacteria need to be presented in Introduction.
Lines 93-102 include this overview, even more convenient as it is focused in the main spoilage microorganisms, that are the ones targeted in the study. Authors consider that enlarging the introduction including useless information for the goal of the study is not necessary and could make the readers loss the focus and interest in the paper. Henceforth, we suggest the reviewer to consider this paragraph as the overview proposed.
As you might understand, the goal of our review comments and suggestions is to reduce weakness (by addressing obvious limitations) in the manuscript, but also to support your main findings. So, no need to repeatedly advocate that aim are microbiota and not the shelf-life, we see that, and we hope you would accept these benevolent suggestions.
We understand perfectly your words and we really thank you for all your efforts done in order to improve the paper. Your suggestions have made us reconsider certain aspects and even including data we have not included as we did not consider them so important. Your invaluable help has made us change our considerations about them and their inclusion has improved substantially the paper. We really thank you for your commitment.
